# The conserved NxNNWHW motif in Aha-type co-chaperones modulates the kinetics of Hsp90 ATPase stimulation

Rebecca Mercier[1], Annemarie Wolmarans[1], Jonathan Schubert[2], Hannes Neuweiler [2], Jill L. Johnson [3] & Paul LaPointe[1]

Hsp90 is a dimeric molecular chaperone that is essential for the folding and activation of hundreds of client proteins. Co-chaperone proteins regulate the ATP-driven Hsp90 client activation cycle. Aha-type co-chaperones are the most potent stimulators of the Hsp90 ATPase activity but the relationship between ATPase regulation and in vivo activity is poorly understood. We report here that the most strongly conserved region of Aha-type co-chaperones, the *N* terminal NxNNWHW motif, modulates the apparent affinity of Hsp90 for nucleotide substrates. The ability of yeast Aha-type co-chaperones to act in vivo is ablated when the *N* terminal NxNNWHW motif is removed. This work suggests that nucleotide exchange during the Hsp90 functional cycle may be more important than rate of catalysis.

[1] Department of Cell Biology, Faculty of Medicine and Dentistry, University of Alberta, Edmonton, AB T6G 2H7, Canada. [2] Department of Biotechnology and Biophysics, University of Würzburg, Würzburg 97074, Germany. [3] Department of Biological Sciences and the Center for Reproductive Biology, University of Idaho, Moscow, ID 83844, USA. Correspondence and requests for materials should be addressed to P.L. (email: paul.lapointe@ualberta.ca)

The 90 kDa heat shock protein (Hsp90) is a dimeric molecular chaperone that promotes the folding and maturation of a large but specific group of substrates called client proteins[1,2]. Client activation during the Hsp90 functional cycle is regulated by a cohort of proteins called co-chaperones[3–13]. Co-chaperones regulate conformational transitions in Hsp90, ATP binding and hydrolysis, as well as client interaction[14,15]. How the Hsp90 functional cycle is regulated in the context of client maturation is poorly understood but it is clear that ATP hydrolysis is critical for efficient client maturation by Hsp90[16,17].

The importance of the Hsp90 ATPase activity has drawn a great deal of attention to the co-chaperones that regulate it. The activator of Hsp90 ATPase, Aha1, is the most potent stimulator of the Hsp90 ATPase activity identified to date[18,19]. Modulating Aha1 levels, and presumably the Hsp90 ATPase activity, has been shown to alter the folding of the cystic transmembrane conductance regulator (CFTR) and its export from the ER[20], kinase activation[21–23], and the activity of other clients[21,22,24]. The cellular activity of Hsp90 appears to be influenced by the relative expression levels of Aha1 and other co-chaperones which are normally far less abundant than the chaperone itself[25,26].

Hsp90 is highly conserved with yeast and human Hsp90 possessing ~60% identity. Co-chaperone proteins are not nearly as well conserved at the level of primary sequence but many are functionally interchangeable between yeast and humans[27–29]. The Aha-type co-chaperones are among the more weakly conserved proteins with the yeast Aha1p and human Ahsa1 sharing only 23% identity but they stimulate the ATPase activity of Hsp90 in a similar manner[30]. Presumably their functional conservation is linked to the sequences that they share. Certainly this is true for the highly conserved RKxK motif which is not required for interaction with Hsp90 but is necessary for robust ATPase stimulation[19]. Curiously, the most strongly conserved region in Aha-type co-chaperones is in the N terminal domain and is defined by the NxNNWHW motif (Fig. 1). This sequence is located in the first 11 amino acids of Aha1p: a region that is not resolved in the co-crystal structure of the Aha1p N domain in complex with the Hsp90 middle domain[19]. Canonical Aha1p is comprised of two domains joined by a flexible linker and yeast possess a smaller, related co-chaperone called Hch1p that lacks the C terminal domain (Fig. 2a). The amino acid sequence of Hch1p is ~35% identical to the Aha1p N domain and it shares both the RKxK and NxNNWHW motifs[19,30].

ATP hydrolysis by Hsp90 occurs in the context of a cycle involving extensive conformational rearrangements[31,32]. Hsp90 is comprised of an N terminal nucleotide-binding domain, joined to a middle domain by a charged linker, which is in turn connected to a C terminal domain that is primarily responsible for dimerization[33]. These different domains are rearranged both in the context of the individual protomers, as well as with respect to the dimer during ATP hydrolysis[34]. Aha1p interacts with the Hsp90 dimer in an anti-parallel fashion where the Aha1p N domain interacts with the middle domain of one Hsp90 subunit and the Aha1p C domain interacts with the Hsp90

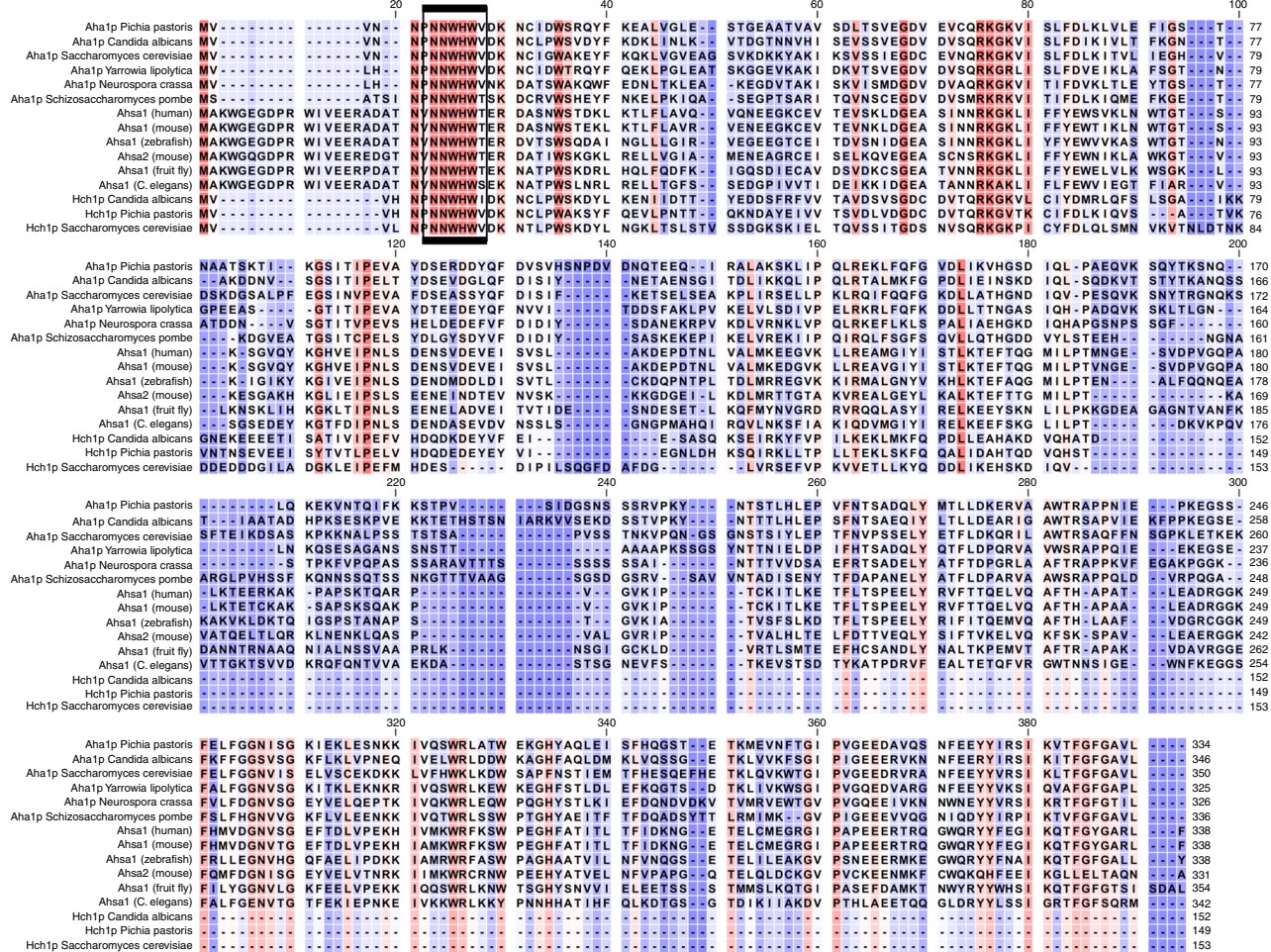

**Fig. 1** Alignment of Aha-type co-chaperones. Alignment of Aha1p and Hch1p homologs shows that the NxNNWHW motif is strongly conserved across species

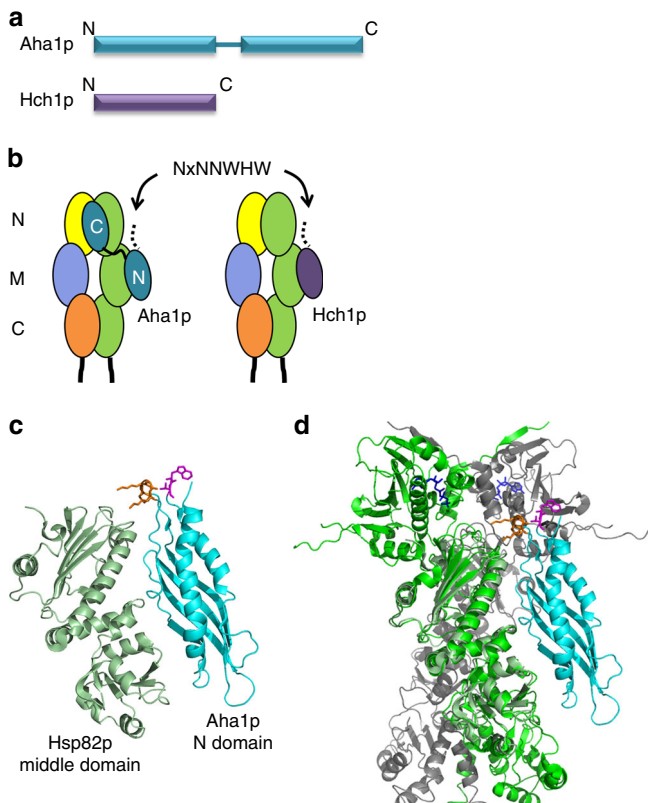

**Fig. 2** Schematic of co-chaperones Aha1p and Hch1p and their interaction with Hsp90. **a** Hch1p corresponds to the N terminal domain of Aha1p which is connected to the C domain by a flexible linker. **b** Hch1p and the Aha1p N domain interact with the middle domain of Hsp90. The Aha1p C domain interacts with the N terminal domains of Hsp90. **c** A model of the complex between the Hsp90 middle domain (light green) and the Aha1p N domain (cyan) (1USV[19]). The RKxK motif is colored in orange and residues Trp11 and Val12 (magenta) indicate where the N terminal NxNNWHW motif would be present (it is unstructured in 1USV). **d** The structure shown in **c** is aligned to the full length, closed Hsp90 dimer structure (2CG9[33]) with the two Sba1p subunits masked. ATP is depicted in blue wireframe. The NxNNWHW motif is predicted to be oriented towards the Hsp90 N domains

N terminal domains after they have come together in the closed conformation[18,19,35,36] (Fig. 2b). Hch1p interacts with the middle domain of Hsp90 in a manner similar to the Aha1p N domain[7,30,37,38].

Aha1p and Hch1p are important regulators of Hsp90 activity but the manner in which these co-chaperones function in vivo and the biological significance of their ATPase stimulation activities are poorly understood[7,9,13,18,19,30,35,36,39]. The NxNNWHW motif is predicted to extend towards the N terminus of the Hsp90 dimer based on a co-crystal structure of the Aha1p N domain and the Hsp90 middle domain[19,33] (Fig. 2c, d). This proximity raises the possibility that the NxNNWHW motif could regulate conformational dynamics associated with the Hsp90 N domain and ATPase activity. In this study, we investigated the importance of the NxNNWHW motif in the biological and biochemical activities of Hch1p and Aha1p.

We report here that the NxNNWHW motif is required for the function of both Hch1p and Aha1p in vivo. Furthermore, the NxNNWHW motif is also required for optimal Hsp90 ATPase stimulation by both co-chaperones, but not for co-operative displacement of Sti1p by Aha1p and Cpr6p in vitro. Strikingly,

the impairment in steady-state-stimulated ATPase activity we observe is not accompanied by an impairment in the rate of conformational transitions en route to the catalytically active conformation leading us to speculate that the NxNNWHW motif could be regulating Hsp90 in part by mediating nucleotide exchange.

## Results

**The NxNNWHW motif is required for optimal ATPase stimulation.** Previous work has shown that the N terminal 11 amino acids harboring the NxNNWHW motif are not important for Aha1p-mediated Hsp90 ATPase stimulation[19]. However, these experiments were carried out with N terminally 6xHis-tagged Aha1p constructs. We wondered if a potential role for the NxNNWHW motif was being masked by the N terminal 6xHis-tag. To explore this possibility we constructed two sets of Aha1p deletion mutants lacking the N terminal 11 amino acids encompassing the NxNNWHW motif (Aha1p$^{\Delta 11}$). One set had N terminal 6xHis-tags and the second had C terminal 6xHis-tags (Fig. 3a). We tested these co-chaperones in ATPase assays with Hsp82p and an enzymatic ATP regenerating system[7,17,30,40,41]. This assay is used to measure $V_{max}$ under steady-state conditions with a fixed (and constant) concentration of ATP. Consistent with previous work, we found that N terminally 6xHis-tagged Aha1p and Aha1p$^{\Delta 11}$ stimulated Hsp82p to a similar degree (Fig. 3b). However, constructs harboring C terminal 6xHis-tags gave different results. Here, Aha1p$^{\Delta 11}$ stimulated the ATPase activity of Hsp82p to a lesser degree than wildtype Aha1p (Fig. 3b). Surprisingly, C terminally 6xHis-tagged Aha1p stimulated ATPase activity to a far greater degree than Aha1p harboring an N terminal 6xHis-tag. This suggests that wildtype Aha1p function is impaired by the N terminal 6xHis-tag. Consequently, we used only C terminally tagged co-chaperone constructs in all our subsequent experiments. It is interesting to note that the apparent affinity of Aha1p$^{\Delta 11}$ for Hsp82p was higher than that of wildtype Aha1p ($0.44 \pm 0.03$ μM compared to $1.24 \pm 0.06$ μM) suggesting that the loss of the NxNNWHW motif stabilizes the interaction with the chaperone.

We constructed C terminally 6xHis-tagged wildtype Hch1p and Hch1p$^{\Delta 11}$ for testing in our ATPase-stimulation assay as well. As we observed with Aha1p, the NxNNWHW motif is important for ATPase stimulation by Hch1p (Fig. 3c). Altogether, the maximal stimulated rate of ATPase activity was ~2.8 fold lower with Aha1p$^{\Delta 11}$ compared to Aha1p, and ~1.7 fold lower with Hch1p$^{\Delta 11}$ than with Hch1p. These data show that the NxNNWHW motif is required for maximal ATPase stimulation by both Hch1p and Aha1p. We also tested Aha1p$^N$ and Aha1p$^{N-\Delta 11}$ in ATPase stimulation assays. Unlike the results we obtained with Hch1p, deletion of the 11 amino acids harboring the NxNNWHW motif did not impair ATPase stimulation by the Aha1p N domain on its own (Fig. 3d). Importantly, we verified that deletion of the NxNNWHW motif did not alter the folding of either Aha1p or Hch1p using a ThermoFluor thermal shift assay[42,43]. The melting curves for both Aha1p$^{\Delta 11}$ and Hch1p$^{\Delta 11}$ were identical to those we obtained for their wildtype counterparts. The melting temperature ($T_m$) of Aha1p and Aha1p$^{\Delta 11}$ were $63.6 \pm 0.9$ and $63.3 \pm 0.9$ °C, respectively. The $T_m$ of Hch1p and Hch1p$^{\Delta 11}$ were $51.5 \pm 2.0$ and $52.6 \pm 1.8$ °C, respectively ($n = 9$).

Aha1p, but not Hch1p, can efficiently displace Sti1p from Hsp90 in concert with Cpr6p[9,37]. We wondered if the NxNNWHW motif might be required for co-chaperone switching. To test this, we reconstituted an ATPase cycling reaction containing Hsp90, Sti1p, Aha1p, and Cpr6p[37]. Sti1p is a strong inhibitor of the Hsp90 ATPase activity and of Aha1p binding to

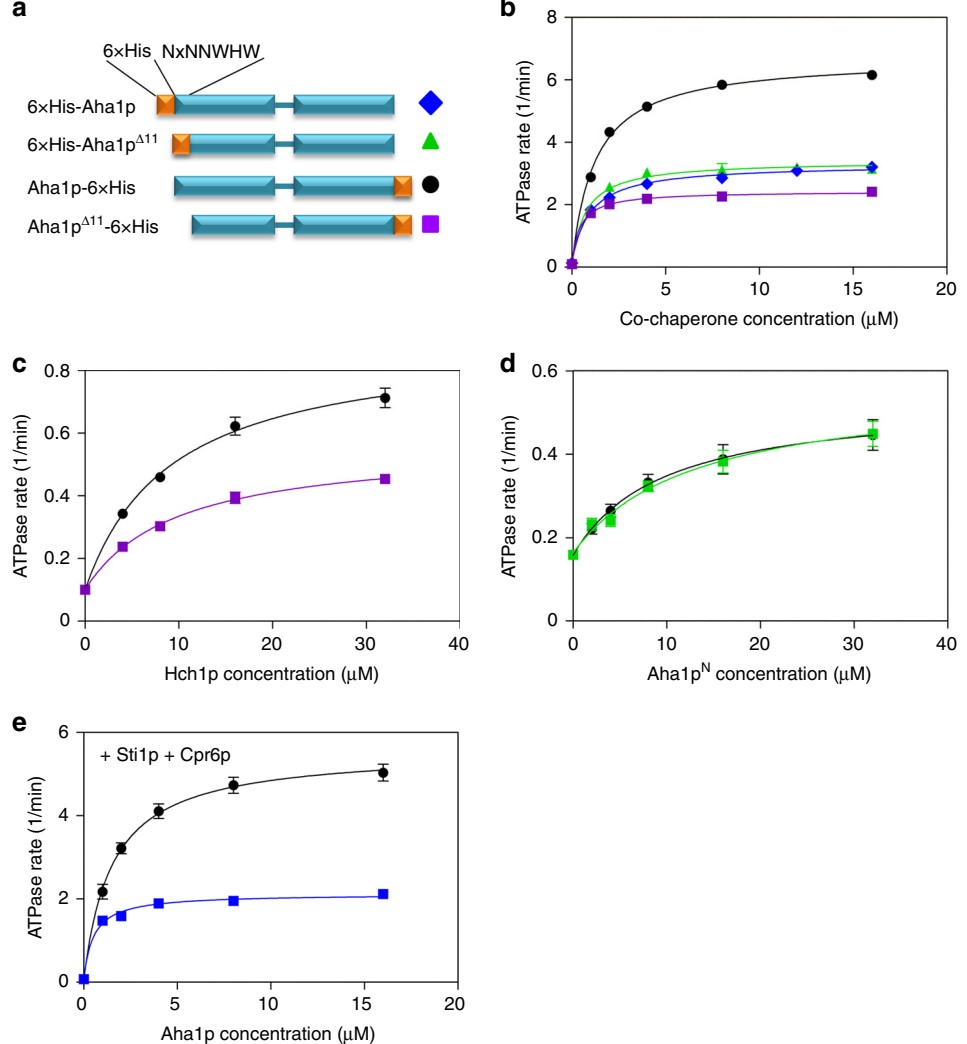

**Fig. 3** Hsp90 ATPase stimulation by Aha1p and Hch1p. **a** Schematic of Aha1p constructs harboring N and C terminal 6xHis-tags. **b** Stimulation of the Hsp82p ATPase activity by increasing concentrations of Aha1p (N terminal 6xHis-tag—blue diamonds; C terminal 6xHis-tag—black circles) and Aha1p$^{\Delta 11}$ (N terminal 6xHis-tag—green triangles; C terminal 6xHis-tag—purple squares). Reactions contained 1 μM Hsp82p and indicated concentration of co-chaperone ($n = 3$). **c** Stimulation of the Hsp82p ATPase activity by increasing concentrations of Hch1p (black circles) and Hch1p$^{\Delta 11}$ (purple squares). Reactions contained 4 μM Hsp82p and indicated concentration of co-chaperone ($n = 3$). **d** Stimulation of the Hsp82p ATPase activity by increasing concentrations of Aha1p$^{N}$ (black circles) and Aha1p$^{N-\Delta 11}$ (green squares). Reactions contained 4 μM Hsp82p and indicated concentration of co-chaperone ($n = 3$). **e** Stimulation of the Sti1p-inhibited Hsp82p ATPase activity by increasing concentrations of Aha1p (black circles) and Aha1p$^{\Delta 11}$ (blue squares) in the presence of Cpr6p. Reactions contained 1 μM Hsp82p, 4 μM Sti1p, and 4 μM Cpr6p ($n = 3$). Error bars show standard error of the mean

Hsp90[8]. Consistent with previous reports[37], robust Hsp90 ATPase stimulation by Aha1p was observed in the presence of Sti1p and Cpr6p (Fig. 3e). Interestingly, Aha1p$^{\Delta 11}$ was also able to stimulate Hsp90 ATPase activity in the presence of Sti1p and Cpr6p suggesting that the NxNNWHW motif is not required for co-chaperone switching in vitro.

**The NxNNWHW motif is required for Hch1p action in cells.** The importance of the N terminal motif for in vivo function of Aha-type co-chaperones has never been investigated. To address this for Hch1p we employed two different yeast assays that we and other have used extensively in other studies. First, we over-expressed either Hch1p or Hch1p$^{\Delta 11}$ in yeast harboring Hsp82p$^{E381K}$ as the sole source of Hsp90 in the cell. This mutant confers slow and temperature-sensitive growth to yeast that is rescued when Hch1p, but not Aha1p, is overexpressed[7,30,44]. Hch1p overexpression rescued growth of yeast expressing

Hsp82p$^{E381K}$ but overexpression of Hch1p$^{\Delta 11}$ did not (Fig. 4a). Both Hch1p and Hch1p$^{\Delta 11}$ were expressed in these strains and were also soluble (Fig. 4b). Second, we measured the cellular sensitivity to the Hsp90 inhibitor, NVP-AUY922, of yeast over-expressing Hch1p or Hch1p$^{\Delta 11}$. Overexpression of Hch1p, but not Aha1p hypersensitizes yeast to Hsp90 inhibitors like NVP-AUY922[7,30]. This experiment revealed that deletion of the NxNNWHW motif in Hch1p eliminated the ability to induce hypersensitivity to NVP-AUY922 (Fig. 4c). Again, both Hch1p and Hch1p$^{\Delta 11}$ were expressed and soluble in lysates from these strains (Fig. 4d).

**The NxNNWHW motif is required for Aha1p action in cells.** We next wondered if the NxNNWHW motif would be required for the in vivo function of Aha1p as well. We identified a mutant of Hsc82p in a screen for *HSC82* mutants that were specifically affected by co-chaperone deletions. Yeast expressing Hsc82p$^{S25P}$

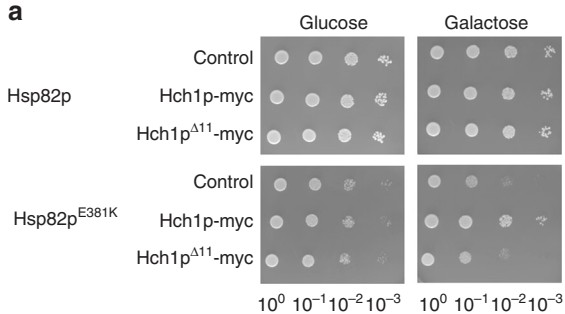

**a**

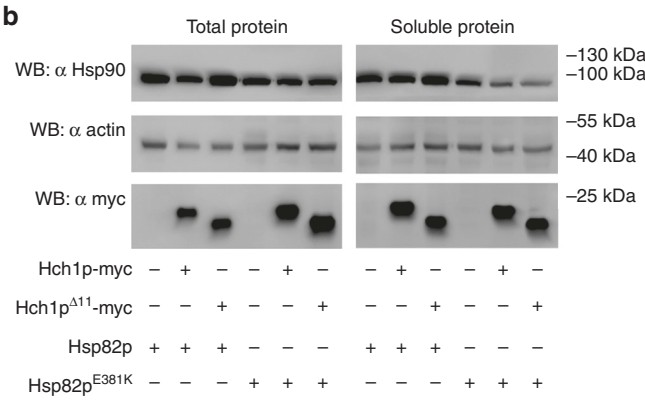

**b**

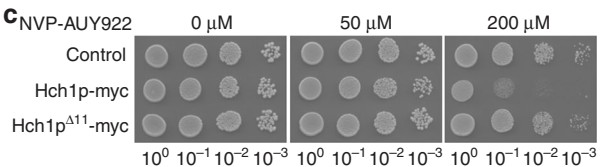

**c**

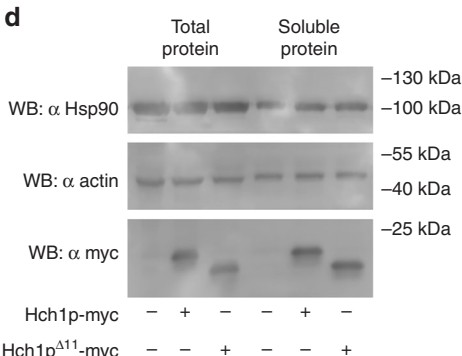

**d**

**Fig. 4** The NxNNWHW motif is required for Hch1p function in vivo. **a** Overexpression of myc-tagged Hch1p, but not Hch1p$^{\Delta11}$, rescues growth defects of yeast expressing Hsp82p$^{E381K}$. Yeast expressing wildtype Hsp82p (ip82a) or Hsp82p$^{E381K}$ (iE381Ka) and harboring expression plasmids encoding the indicated co-chaperones were grown overnight at 30 °C in SC media lacking uracil (SC-Ura) and containing 2% raffinose and then diluted to $1 \times 10^8$ cells per milliliter. We prepared 10-fold serial dilutions and spotted 10 μL aliquots on SC-Ura agar plates supplemented with either 2% glucose or galactose. Plates were incubated for 2 or 3 days for ip82a and iE381Ka strains, respectively, at 30 °C. **b** Western blot of total lysates and the soluble protein fraction from the yeast strains shown in **a** were probed with anti-Hsp90, anti-actin, and anti-myc antibodies. **c** Overexpression of myc-tagged Hch1p, but not Hch1p$^{\Delta11}$, confers hypersensitivity to NVP-AUY922 in yeast. Yeast expressing wildtype Hsp82p (ip82a) and harboring expression plasmids encoding the indicated co-chaperones were grown overnight at 30 °C in YPD supplemented with 200 mg/L G418 and then diluted to $1 \times 10^8$ cells per milliliter. We prepared 10-fold serial dilutions and spotted 10 μL aliquots on YPD agar plates supplemented with 200 mg/L G418 and the indicated concentrations of NVP-AUY922. Plates were incubated for 2 days at 30 °C. **d** Western blot of total lysates and the soluble protein fraction from the yeast strains shown in **c** were probed with anti-Hsp90, anti-actin, and anti-myc antibodies. Representative results of three independent experiments are shown

phospho-tyrosine. Consistent with what we observed in our growth assays, the overexpression of Aha1p, but not Aha1p$^{\Delta11}$, in the Hsc82p$^{S25P}$ background resulted in the accumulation of phospho-tyrosine when v-src expression was induced (Fig. 5d). Interestingly, v-src accumulated to a comparable degree regardless of Aha1p or Aha1p$^{\Delta11}$ expression but tyrosine kinase activity was only restored when Aha1p was overexpressed.

**The S25P mutation impairs ATPase stimulation by Aha1p.** We wondered if the S25P mutation conferred an ATPase defect to Hsp82p. We expressed and purified Hsp82p$^{S25P}$ for analysis in our ATPase assays. The intrinsic ATPase activity of Hsp82p$^{S25P}$ was comparable to that of wildtype Hsp82p (Fig. 6a). In fact, Hsp82p$^{S25P}$ displayed a small, but statistically significant, increase in intrinsic ATPase activity compared to wildtype Hsp82p. However, stimulation of Hsp82p$^{S25P}$ by Aha1p was vastly reduced compared to wildtype Hsp82p and was almost completely dependent on the presence of the NxNNWHW motif as the Aha1p$^{\Delta11}$ mutant barely stimulated the ATPase rate of Hsp82p$^{S25P}$ (Fig. 6b).

**The NxNNWHW motif modulates apparent affinity for ATP.** To better understand the underlying mechanism for the different stimulated ATPase rates we observed, we measured ATP hydrolysis at different concentrations of ATP. We carried out ATPase assays with Hsp82p on its own, as well as during stimulation by Aha1p or Hch1p, with and without the NxNNWHW motif. We used varying ATP concentrations (12.5, 25, 50, 100, 200, 400, 800, 1600 μM) and analyzed the data (see Methods) to determine the apparent $K_M$ for ATP of Hsp90 (Supplementary Figure 1). The curve fits all had $R^2$ values >0.95 (>0.9 for Hsp82p alone) giving us very high confidence in the apparent $K_M$ values we calculated.

Consistent with previous findings, we observed no significant change in the apparent $K_M$ for ATP Hsp82p on its own or when stimulated with Aha1p[13] (Fig. 7a). Intriguingly, Hch1p resulted in a large increase in apparent $K_M$ for ATP (~3.5 fold) suggesting this co-chaperone reduces the affinity for nucleotide (Fig. 7b). Deletion of the NxNNWHW motif in either Aha1p or Hch1p resulted in a significant decrease in $K_M$ relative to their full-length counterparts.

as the sole source of Hsp90 exhibit temperature-sensitive growth that is worsened by deletion of *AHA1* (Fig. 5a). We used this strain to investigate Aha1p action in the context of the NxNNWHW motif. Overexpression of Aha1p in this strain rescued growth at elevated temperatures (Fig. 5b). However, overexpression of Aha1p$^{\Delta11}$ did not have any effect on the growth of this strain. Of note, both the wildtype and N terminally truncated versions of Aha1p were expressed to comparable levels and were soluble (Fig. 5c). This suggests that the NxNNWHW motif is important for Aha1p function in this strain background.

The S25P mutation has not been studied before so we wondered if this substitution impaired client activation in cells. We used v-src as a marker for Hsp90 client activation with and without overexpression of Aha1p constructs. We employed a system where v-src was expressed from a galactose-inducible promoter and monitored the activation of this kinase by western blotting for the stability of v-src as well as the accumulation of

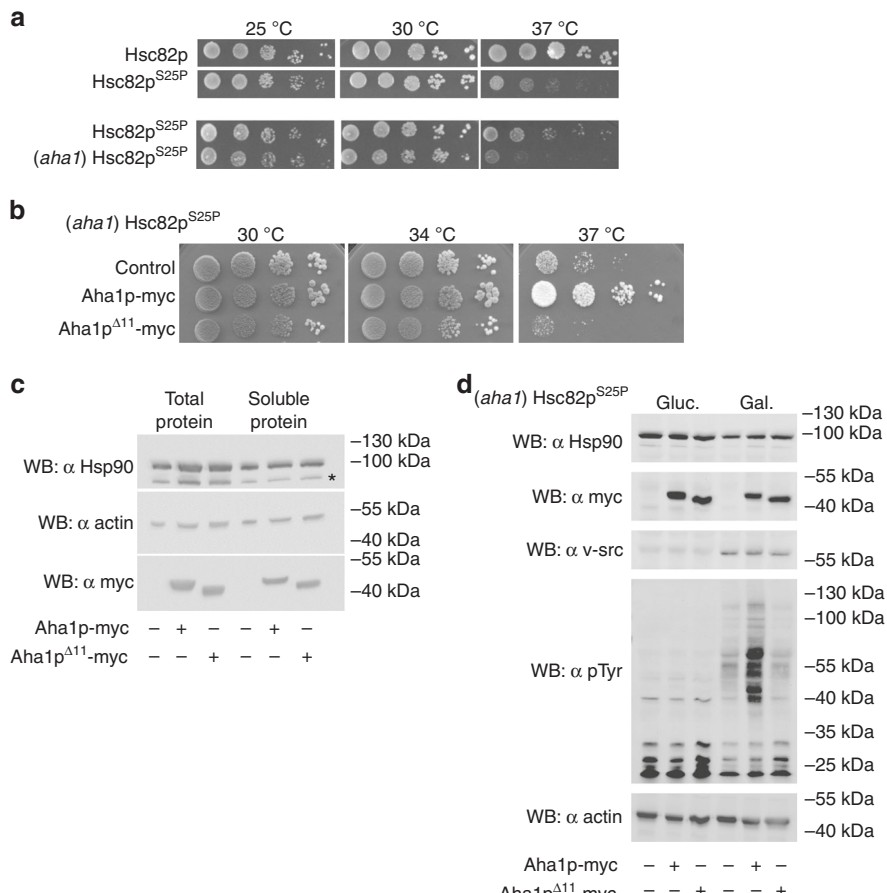

**Fig. 5** The NxNNWHW motif is required for Aha1p function in vivo. **a** Yeast expressing Hsc82p[S25P] as the sole source of Hsp90 in the cell exhibit temperature-sensitive growth. The deletion of AHA1 in yeast expressing Hsc82p[S25P] exacerbates the temperature-sensitive growth defect. **b** Overexpression of myc-tagged Aha1p, but not Aha1p[Δ11], rescues the growth of yeast expressing Hsc82p[S25P]. Yeast expressing Hsc82p[S25P] and harboring expression plasmids encoding the indicated co-chaperones were grown overnight at 30 °C in YPD supplemented with 300 mg/L Hygromycin and then diluted to $1 \times 10^8$ cells per milliliter. We prepared 10-fold serial dilutions and spotted 10 μL aliquots on YPD agar plates supplemented with Hygromycin 300 mg/L. Plates were incubated for 2 days at 30, 34, or 37 °C. **c** Western blot of total lysates and soluble protein extracted from the yeast strains shown in **b** were probed with anti-Hsp90, anti-actin, and anti-myc antibodies. **d** Overexpression of Aha1p, but not Aha1p[Δ11], enhances v-src activation in yeast expressing Hsc82p[S25P]. Yeast expressing Hsc82p[S25P] and harboring expression plasmids encoding the indicated co-chaperones and a galactose inducible v-src expression plasmid were grown overnight at 30 °C in SC-Ura containing 2% raffinose, supplemented with 300 mg/L hygromycin. Cells were diluted to an OD_{600} of 0.5 and grown for an additional 6 h in SC-Ura containing either 2% glucose or galactose, supplemented with hygromycin (300 mg/L). Yeast strains were probed with anti-Hsp90, anti-myc, anti-v-src, anti-phosphotyrosine, and anti-actin antibodies. Representative results of three independent experiments are shown

**The NxNNWHW motif does not influence lid closure.** ATP hydrolysis by Hsp90 is rate-limited by conformational change[34,45,46]. Local conformational changes within the chaperone machinery are kinetically linked and appear to cooperate[34]. Briefly, ATP binding drives closure of the lid over the binding pocket, exchange of N terminal strands between the two protomers of the N terminally dimerized Hsp90, and docking of the N domains with the middle domains of Hsp90. These coordinated events are stimulated when Aha1p binds to Hsp90[34]. Following the structural model (Fig. 2b–d), the NxNNWHW-containing peptide of Aha1p is located in close proximity to the N domain of Hsp90. Presence of this conserved motif is thus likely to influence the N terminal ATP-binding domain of Hsp90, which appears to be reflected in our observed alteration of $K_M$. To explore the mechanistic basis for reduced ATPase stimulation of Hsp90 by Aha1p[Δ11] and the associated reduction of $K_M$ of ATP we measured the rate constants of closure of the lid over the ATP-binding pocket in the N domain using photoinduced electron transfer (PET) fluorescence quenching, as previously described[34].

Previous work demonstrated that Aha1p accelerates closure of the lid and the other associated conformational events. In Michaelis–Menten kinetics, a reduction of $K_M$ can originate from a reduction of rate constant of ATP hydrolysis or from modulation of ATP binding (i.e. from a reduced rate constant of ATP dissociation or increased rate constant of ATP association), or both[47,48]. We hypothesized that the deletion of the NxNNWHW motif would likely affect kinetics of conformational change (i.e. would slow lid closure and therefore reduce the rate constant of ATP hydrolysis by Hsp90). We measured the time constant ($\tau$) of lid closure of yeast Hsp90, which was $\tau = 320 \pm 100$ s (Fig. 8a), in agreement with previous findings[34]. Surprisingly, we found that lid closure was accelerated to an identical degree by Aha1p and Aha1p[Δ11] ($\tau = 14 \pm 2$ s and $\tau = 15 \pm 2$ s, respectively), showing that the decrease in ATPase stimulation we observed with Aha1p[Δ11] was not due to a defect in the acquisition of the catalytically competent state (Fig. 8b). We next wondered how Hch1p (which regulates Hsp90 differently in cells[7,30]) affects kinetics of lid closure and acquisition of the catalytically

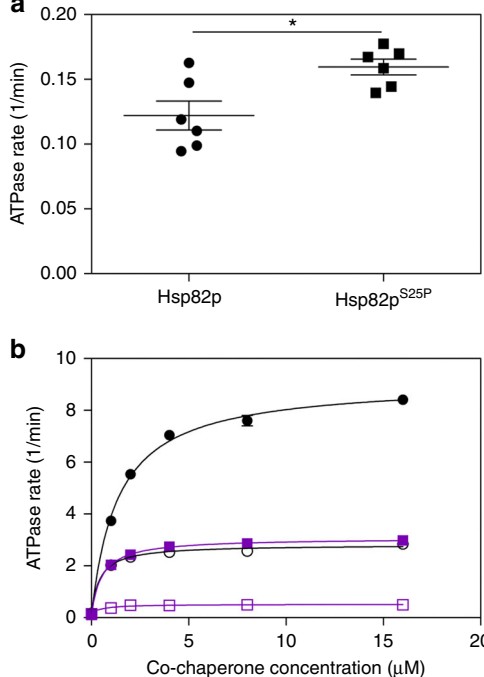

**Fig. 6** Hsp82p$^{S25P}$ ATPase stimulation by Aha1p is dependent on the NxNNWHW motif. **a** The intrinsic ATPase activity of Hsp82p$^{S25P}$ is comparable to that of wildtype Hsp82p. Reactions contained 5 μM Hsp82p or Hsp82p$^{S25P}$ ($n = 6$). Statistical significance (*) was determined using an unpaired $t$-test. Error bars show standard error of the mean. **b** Stimulation of the ATPase activity of wildtype Hsp82p and Hsp82p$^{S25P}$ by increasing concentrations of Aha1p (closed black circles, open black circles, respectively) and Aha1p$^{Δ11}$ (closed purple squares, open purple squares, respectively). Reactions contained 1 μM Hsp82p and indicated concentration of co-chaperone ($n = 3$). Error bars show standard error of the mean

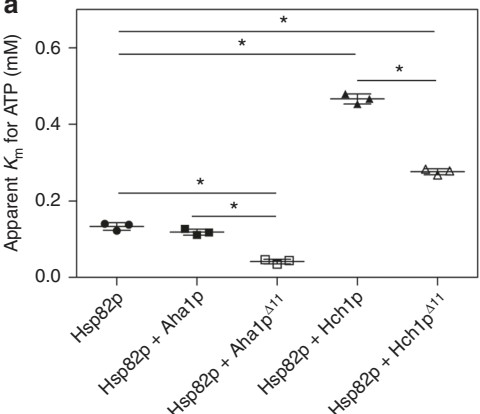

**b**

| Condition | Apparent $K_m$ for ATP |
|---|---|
| Hsp82p alone | 0.133 +/− 0.010 |
| Hsp82p + Aha1p | 0.118 +/− 0.008 |
| Hsp82p + Aha1p$^{Δ11}$ | 0.042 +/− 0.006 |
| Hsp82p + Hch1p | 0.466 +/− 0.013 |
| Hsp82p + Hch1p$^{Δ11}$ | 0.277 +/− 0.008 |

**Fig. 7** The NxNNWHW motif modulates the apparent $K_M$ for ATP of Hsp90. **a** Kinetic analysis was carried out for Hsp82p alone or in the presence of Aha1p, Aha1p$^{Δ11}$, Hch1p, or Hch1p$^{Δ11}$. ATPase reactions were carried out with increasing concentrations of ATP (12.5, 25, 50, 100, 200, 400, 800, 1600 μM) and ATPase rates were analyzed with the Michaelis–Menten non-linear regression function in GraphPad Prism. The curve fits all had $R^2$ values >0.9. The apparent $K_M$ values for three experiments are shown in the scatter plots. Statistical significance (*) was calculated pair-wise with a $t$-test ($n = 3$; $n = 4$ for Hsp82p alone). Error bars show standard deviation. **b** Table showing $K_M$ values plotted in **a**

competent state. We tested Hch1p and Hch1p$^{Δ11}$ in this assay and found, again, that time constants of lid closure were within error, i.e. ($τ = 105 ± 13$ s and $τ = 120 ± 16$ s, respectively) (Fig. 8c). Unchanged kinetics of conformational change lead us to conclude that reduction of $K_M$ of ATP/Hsp90 in Aha-type NxNNWHW deletion mutants compared with wildtype Aha1 originated from tighter binding of nucleotide in the N domain-binding pocket (i.e. from a reduced dissociation constant of bound nucleotide induced by interaction with the NxNNWHW motif). Interestingly, binding of Aha-type co-chaperones with and without the NxNNWHW motif resulted in different fluorescence intensities of the label positioned on the N terminal domain (Fig. 8b, c), which shows that there is indeed an interaction of the NxNNWHW motif with the N terminal domain.

**Loss of the NxNNWHW motif promotes the Hsp90 closed state.** Loss of the NXNNWHW motif did not appear to slow the acquisition of the catalytically competent state of Hsp90 compared to full-length Aha1p in our PET fluorescence measurements. This was surprising since ATPase stimulation of Hsp90 by Aha1p$^{Δ11}$ was impaired at steady state. We speculated that this could be because the NxNNWHW is involved in release of ADP after hydrolysis has occurred. Slower ADP release would result in a slower cycling rate. Sba1p is a co-chaperone that binds to the N terminally dimerized, catalytically competent state of Hsp90[33]. Sba1p does not affect Hsp90 ATPase activity in single turnover reactions but inhibits ATPase activity at steady state[49]. This is because it slows the release of ADP after hydrolysis has occurred.

If Aha1p$^{Δ11}$ promotes the acquisition of the ATPase-competent state but cannot promote ADP dissociation then we would predict the apparent affinity of Sba1p would be higher for Hsp90 in the presence of Aha1p$^{Δ11}$ than in the presence of Aha1p. To test this, we titrated Sba1p into Hsp90 ATPase reactions stimulated with either Aha1p or Aha1p$^{Δ11}$. Strikingly, despite Aha1p$^{Δ11}$ having a higher apparent affinity for Hsp82p than Aha1p, Sba1p had a far higher apparent affinity for Hsp82p that was stimulated by Aha1p$^{Δ11}$ (Fig. 9).

## Discussion

Aha1p and Hch1p share several sequence elements that are important for regulating the ATPase activity of Hsp90 in vitro as well as Hsp90 function in vivo[7,13,19,30,44]. We report here that the strongly conserved NxNNWHW motif, harbored in the first 11 amino acids of Aha1p and Hch1p (and all Aha-type co-chaperones) (Fig. 1), is critical for the in vivo function of both co-chaperones. This region is not fully resolved in the co-crystal structure of the Hsp90 middle domain and the Aha1p N domain[19]. However, its location can be inferred from the location and orientation of the Aha1p N domain in complex with the Hsp90 middle domain[19] in the context of the structure of full-length Hsp90[33,35,36]. The NxNNWHW motif extends towards the N terminal Hsp90 ATPase domains (Fig. 2). This is consistent with our enzymatic data which shows that the NxNNWHW motif is important for ATPase stimulation by both co-chaperones (Fig. 3).

Several intra-subunit and inter-subunit rearrangements occur during the Hsp90 ATP hydrolysis cycle. Access to the nucleotide-binding pocket is controlled by a lid (amino acids 98–121)[50]. This lid is in the open position in the absence of nucleotide but

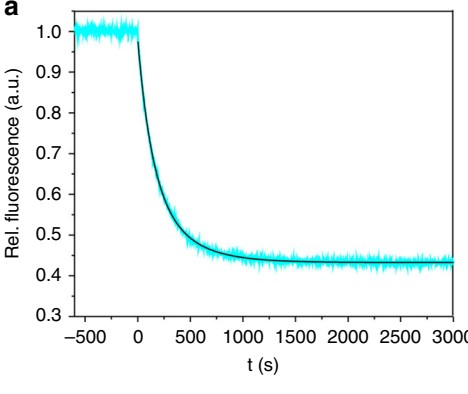

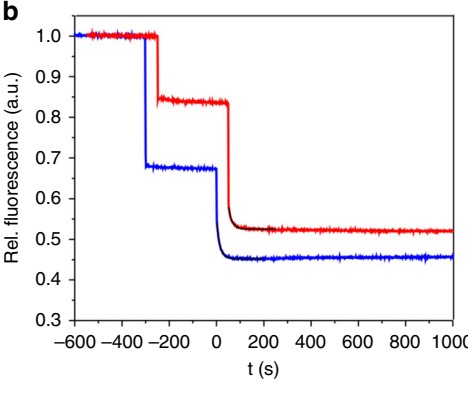

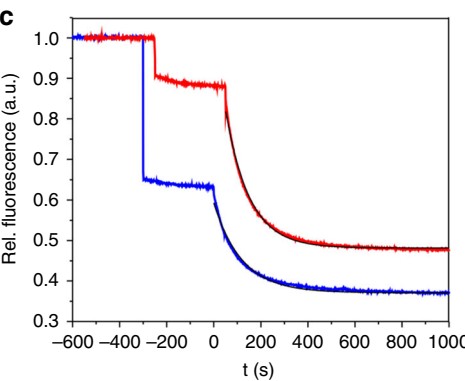

**Fig. 8** Kinetics of lid closure of Hsp90 and its modulation by Aha-type co-chaperones. **a** Time course of PET fluorescence quenching of fluorescently modified Hsp90 mutant S51C-A110W (cyan). The non-hydrolyzable ATP analog AMP-PNP, which traps Hsp90 in the closed-clamp conformation, was added at time $t = 0$. The black line is a fit to a bi-exponential decay function. **b** Same experiment as shown in **a** but Aha1p (blue line) or Aha1p$^{\Delta 11}$ (red line) was added 300 s prior to addition of AMP-PNP. Black lines are fits to exponential decay functions. **c** Same experiment as shown in **a** but Hch1p (blue line) or Hch1p$^{\Delta 11}$ (red line) was added 300 s prior to addition of AMP-PNP. Black lines are fits to exponential decay functions. Data sets of Aha1p$^{\Delta 11}$ and of Hch1p$^{\Delta 11}$ are offset along the x-axis for reasons of clarity

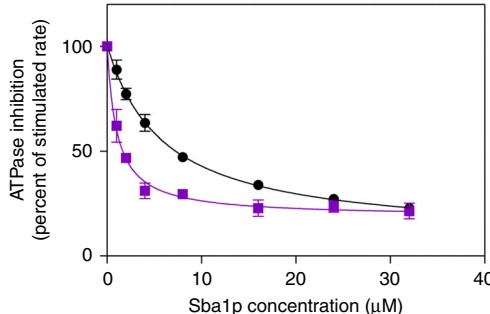

**Fig. 9** Sba1 has a higher apparent affinity for Hsp82p in the presence of Aha1p$^{\Delta 11}$. Increasing concentrations of Sba1p were titrated into reactions containing Hsp82p and either Aha1p (black circles) or Aha1p$^{\Delta 11}$ (purple squares) ($n = 4$). Reactions contained 1 μM Hsp82p, 5 μM of either Aha1p or Aha1p$^{\Delta 11}$, and the indicated concentration of Sba1p. Error bars show standard deviation

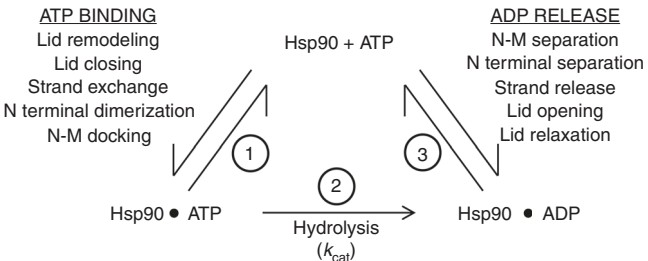

**Fig. 10** A simplified schematic of the Hsp90 ATPase cycle. ATP binding is accompanied by several conformational rearrangements that must occur for Hsp90 to attain its catalytic state (1). Once ATP hydrolysis occurs (2), these conformational changes must be reversed for ADP to be released and another round of ATP binding and hydrolysis to be initiated (3)

undergoes a rapid remodeling event followed by slow closure upon ATP binding[34]. This traps ATP in the binding pocket and renders it committed to hydrolysis[51]. Lid closure is accompanied by dimerization of the N terminal domains and exchange of a beta strand (amino acids 1–8) between the two protomers[9,34,46,50,52]. This conformation is thought to be catalytically active. The conformational changes that occur after ATP-binding comprise the rate limiting step(s) of the ATPase cycle[51]. Aha1p is well known to activate Hsp90 ATPase activity by

accelerating these steps[9,34,46,53,54]. After ATP hydrolysis occurs, the molecular clamp must open in order for ADP and inorganic phosphate to be released and a new cycle of ATP binding and hydrolysis to be initiated (Fig. 10). However, the order and kinetics of events associated with ADP release after ATP hydrolysis are not well understood, especially in the context of stimulation by co-chaperones-like Aha1p and Hch1p, and are extremely difficult to measure directly.

Deletion of the NxNNWHW motif results in a clear defect in ATPase stimulation of Hsp90 (Fig. 3). A defect in the acquisition of the catalytic state seems like the simplest explanation for the reduction in $V_{max}$ we observe with Aha1p$^{\Delta 11}$. Such a defect could be in the ability of Aha1p to mobilize the catalytic loop when the NxNNWHW motif is deleted. However, deletion of the NxNNWHW motif did not impair Hsp90 ATPase stimulation by the Aha1p N terminal domain on its own (Fig. 3d) suggesting that both Aha1$^N$ and Aha1$^{N-\Delta 11}$ are equally able to remodel the catalytic loop. The ATPase stimulation defect we observe upon deletion of the NxNNWHW motif in full length Aha1p appears to be dependent on the presence of the Aha1p C terminal domain (Fig. 3b). Thus, in Aha1p, the NxNNWHW motif and the C terminal domain act cooperatively to promote full stimulation of Hsp90 at steady state. The C domain of Aha1p is required for robust ATPase stimulation (compared to the Aha1p N domain alone) not because of a role in catalytic loop remodeling, but rather because of its interaction with the Hsp90 N domains and acquisition of the N terminally dimerized state[35–37]. However, co-chaperone switching (Sti1p displacement) was equally efficient with either Aha1p$^{\Delta 11}$ or Aha1p (Fig. 3e). In addition, Sba1p, which

is a sensor for the N terminally dimerized state of Hsp90[55,56], actually bound with a higher apparent affinity to Hsp90 in complex with Aha1p$^{\Delta 11}$ than with Aha1p (Fig. 9). This observation may provide the strongest clue regarding the role of Aha1p in nucleotide release because Sba1p does not inhibit ATPase hydrolysis in single turnover reactions but robustly inhibits at steady state[49]. Sba1p stabilizes the N terminally dimerized state of Hsp90 and slows ADP release, after which Sba1p can no longer bind. A higher apparent Sba1p-binding affinity is consistent with ADP release being slower in the presence of Aha1p$^{\Delta 11}$ than wildtype Aha1p. Of course we must acknowledge that the NxNNWHW motif may pose a steric barrier to Sba1p binding as well. In all these experiments we were unable to identify a defect in the ability of the Aha1p$^{\Delta 11}$ mutant to promote the catalytically active, N terminally dimerized state. In fact, direct measurement of lid closure revealed identical rates with Aha1p and Aha1p$^{\Delta 11}$ (Fig. 8b). What then could account for the reduction in $V_{max}$ in our cycling ATPase assays with Aha1p$^{\Delta 11}$? We observed an increase in apparent affinity for nucleotide in ATPase assays with both Aha1p$^{\Delta 11}$ and Hch1p$^{\Delta 11}$ compared to their full length counterparts (Fig. 7). Since $k_{on}$ is diffusion controlled and closing of the clamp was unaltered by the deletion of NxNNWHW motif, we propose that the rate of ADP release (after hydrolysis) is slower in the presence of Aha1p$^{\Delta 11}$ compared to Aha1p. Certainly, dissociation of ADP would be rate limiting for a new cycle of ATP hydrolysis to be initiated at steady state (Fig. 10).

Deletion of the NxNNWHW motif affects Aha1p and Hch1p in a similar manner despite the different roles these co-chaperones play in regulating Hsp90 biology in yeast[7,30]. Aha1p is known to elicit a partially closed lid conformation that one would expect to make ATP-trapping more efficient and lower the $K_M$ for ATP[5,9,46]. However, in agreement with previous studies, we did not observe a change in the apparent $K_M$ for ATP in Aha1p-stimulated reactions compared to reactions containing Hsp90 alone[13]. There was no difference between stimulated ATPase rates in reactions containing the Aha1p N domain or a variant lacking the NxNNWHW motif in our hands (Fig. 3d). This suggests that, at least for Aha1p, the NxNNWHW motif acts cooperatively with the C domain. In the case of Hch1p, the pronounced effect on the apparent $K_M$ of Hsp90 for ATP compared to Aha1p may explain the ability of Hch1p to regulate cellular sensitivity to Hsp90 inhibitors[7]. Overexpression of Hch1p, but not Aha1p, confers sensitivity to Hsp90 inhibitors in yeast[7]. If Hch1p stimulates Hsp90 ATPase activity in part by promoting nucleotide exchange, this may allow greater access for ATP-competitive Hsp90 inhibitors to the ATP-binding pocket. The deletion of the NxNNWHW motif in Hch1p dramatically lowered the ATPase rate and $K_M$ for ATP of Hsp90 and ablated the ability of Hch1p to sensitize yeast to Hsp90 inhibitors.

Deletion of the NxNNWHW in either Hch1p or Aha1p eliminates the in vivo activity of these co-chaperones while only affecting nucleotide exchange in vitro. That the deletion of the NxNNWHW motif does not affect the ability of Aha1p$^{\Delta 11}$ and Hch1p$^{\Delta 11}$ to promote the acquisition of the catalytically active state but eliminates their biological activity suggests that nucleotide exchange (or release) is the critical function of these co-chaperones. The precise manner in which ATP hydrolysis and nucleotide exchange fit into the client activation cycle has not been elucidated. Recent work has even called the significance of ATP hydrolysis into question while reaffirming the importance of nucleotide binding for Hsp90 function[39]. Additionally, there is also some evidence that mixed nucleotide-bound states (one protomer bound to ADP and one bound to ATP) may also be functionally significant[57]. Focusing on other elements of the functional cycle, such as exchange dynamics during cycling, may reveal new insight into how client activation is regulated.

## Methods

**Yeast strains and plasmids**. Yeast galactose-inducible plasmids were constructed by amplifying HCH1 and AHA1 coding sequences (wildtype (primer 13 for AHA1; primer 17 for HCH1) or with the 11 amino acids harboring the NxNNWHW motif deleted (Δ11) (primer 573 for AHA1; primer 571 for HCH1)) by PCR to have upstream BamHI and downstream SacI (primer 90 for HCH1) or NotI (primer 126 for AHA1) sites for cloning into pRS416GAL (for wildtype Aha1p and Hch1p) or pRS426GAL (for Aha1p$^{\Delta 11}$ and Hch1p$^{\Delta 11}$)[58]. The HCH1 and AHA1 coding sequences (wildtype (primer 13 for AHA1; primer 17 for HCH1) or Δ11 variant (primer 573 for AHA1; primer 571 for HCH1)) were amplified by PCR with primers designed to introduce a BamHI site at the 5′ end and a myc tag and XhoI site at the 3′ end (primer 16 for AHA1; primer 20 for HCH1). These PCR products were then digested with BamHI and XhoI and cloned into similarly cut p41KanTEF[7]. Site-directed mutagenesis for Hsp82(S25P) was carried out using QuikChange™ mutagenesis according to the manufacturers protocol (Agilent) using primers 672 and 673. The coding sequences contained in all mutagenized plasmids were verified by sequencing. We constructed our p42HygGPD vector by digesting the GPD and CYC1 terminator fragment of p414GPD[59] with SacI and KpnI and ligating into similarly cut pRS42H[60]. The HCH1 and AHA1 (wildtype or Δ11 variant) coding sequences were cut from p41KanTEF plasmids using BamHI and XhoI and cloned into p42HgyGPD cut BamHI and SalI to yield p42HygGPDHch1myc, p42HygGPDHch1$^{\Delta 11}$myc, p42HygGPDAha1myc, and p42HygGPDAha1$^{\Delta 11}$myc. The yeast strains ip82a, iE381K, and iTHisHsp82p were derived from ΔPCLDa (kindly provided by Dr. Susan Lindquist)[7,30,61]. In the case of yeast strains transformed with our p41KanTEF or p42HygGPD plasmids, transformants were selected on YPD supplemented with G418 (200 mg/L) (Goldbio, USA) or Hygromycin (300 mg/L) (Goldbio, USA) respectively.

Bacterial expression vectors encoding N terminally His-tagged Hch1p and Aha1p lacking the first 11 amino acids (Δ11) were constructed in a similar manner to those encoding wildtype Aha1p and Hch1p[7,30,37]. We amplified the HCH1 (primers 601/27) and AHA1 (primer 572/29) coding sequences were amplified by PCR to introduce an NdeI site at the 5′ end and a BamHI site at the 3′ end. These products were then digested with NdeI and BamHI and ligated into similarly cut pET11dHis. Bacterial expression vectors encoding C terminally His-tagged Hch1p and Aha1p (wildtype (primers 657/659 for AHA1; primer 654/656 for HCH1) and lacking the first 11 amino acids (Δ11) (primer 658/659 for AHA1; primer 655/656 for HCH1)) were constructed by amplifying HCH1 and AHA1 coding sequences by PCR to introduce an NcoI (HCH1) or XbaI (AHA1) site at the 5′ and a BamHI site at the 3′ end. These PCR products were digested with NcoI and BamHI (HCH1) or XbaI and BamHI (AHA1) and ligated into similarly cut pET11dHis. For C-terminally His-tagged Aha1p$^N$ domain, the same procedure was followed as for Aha1p except primers 657 and 664 were used (primers 658 and 664 for the Δ11 variant). All vectors were sequenced to verify final coding sequences.

The Hsc82p$^{S25P}$ mutant was identified in a yeast screen[62]. Briefly, the coding sequence of HSC82 was subjected to error-prone mutagenesis. The library was screened by transformation into an hsc82hsp82 strain or hsc82hsp82 strain containing deletion of non-essential co-chaperones and selecting colonies that failed to support growth when plated onto 5-FOA at 37 °C. Library plasmids that produced wildtype levels of Hsc82p were rescued and sequenced fully. The pRS316 v-src plasmid was a gift from Dr. David Morgan (University of California)[63] and transformants were selected on SC-uracil. All primers are listed in Supplementary Figure 2.

**Growth assays**. Strains were grown in defined media or YPD (where indicated), with or without G418 (200 mg/L) or hygromycin (300 mg/L), diluted to $1 \times 10^8$ cells per mL and 10-fold serial dilutions were prepared as indicated. Ten microliters of drops were placed on agar plates (YPD or defined, with or without NVP-AUY922 (at indicated concentrations), with or without G418 or hygromycin) and grown for 48 h unless otherwise indicated, at indicated temperatures. Growth assays with yeast expressing Hsc82p$^{S25P}$ were carried out as above in strain JJ816 (hsc82hsp82) and JJ95 (aha1hsc82hsp82)[64].

**v-src activation assay**. Yeast were grown overnight at 30 °C in SC-Ura with 2% raffinose. After overnight growth, cells were diluted to an OD$_{600}$ of 0.5 and grown for an additional 6 h rotating at 30 °C in 5 mL of appropriate media supplemented with 2% glucose, for inhibition of the plasmid, or 2% galactose, for plasmid induction. After 6 h the OD$_{600}$ was measured and five units of cells were harvested for protein extraction and analyzed by western blot.

**Lysate generation and Western blotting**. Yeast were grown overnight at 30 °C in appropriate media. For analysis of total protein content, five OD$_{600}$ units of cells were transferred to a microfuge tube, washed with distilled water and pelleted for processing. Cells were resuspended in 500 μL of distilled water, 90 μL of lysis buffer (2.2 M NaOH, 1 M β-mercaptoethanol, 10 mM PMSF) was added and samples vortexed twice for 30 s. 250 μL of 100% TCA was added and samples were vortexed briefly and then precipitated in a cold microcentrifuge. Pellets were washed twice with acetone, dried and then resuspended in sample buffer for analysis by SDS–PAGE and western blotting. For analysis of soluble protein content, 35 OD$_{600}$ units of cells were resuspended in 1 mL of lysis buffer (50 mM Tris

pH 7.5, 100 mM KCl, 5 mM MgCl$_2$, 20 mM Na$_2$MoO$_4$, 20% glycerol, 5 mM β-mercaptoethanol, HALT EDTA-free protease inhibitor (Thermo scientific)) and added to a 2-mL screw cap tube that was half-filled with 0.5 mm glass beads (Biospec, Bartlesville, OK, USA). Cells were lysed with a Mini-Beadbeater-16 for 3 min, and supernatant was clarified by centrifugation at 20,800 rcf for 10 min. Protein from each supernatant was TCA precipitated and resuspended in sample buffer for analysis by SDS–PAGE and western blotting. Myc-tagged proteins were detected with mouse anti-myc monoclonal antibody[65] (1:100; 4A6 Millipore, catalog number 05-724) and Hsp82p was detected with anti-Hsp90 antibody (1:1000; Anti-Hsp90, Clone K41220A, Stressmarq Biosciences Inc., Victoria, BC, Canada; catalog number SMC-135), v-src was detected with anti-v-src antibody (1:200; clone 327 Sigma-Aldrich; catalog number MABS193), phosphotyrosine levels were detected with anti-phosphotyrosine antibody (1:1000; Stressmarq Biosciences Inc., Victoria, BC, Canada; catalog number SMC-157). Anti-actin antibodies were kindly provided by Dr. Gary Eitzen (1:2000; University of Alberta).

**Genomic analysis.** We performed reciprocal BLAST searches with the Hch1p and Aha1p protein sequences against a large number of published genomes with which predicted proteins sequences exist. Our criteria for classification of proteins as Hch1p-like or Aha1p-like are as follows. Hch1p-like proteins contained one or more of the NxNNWHW and RKxK motifs and D53 but lacked a recognizable C-terminal domain similar to Aha1p[157–350]. Aha1p-like proteins contained one or more of the NxNNWHW and RKxK motifs, and D53 and possessed a recognizable C-terminal domain.

**Protein expression and purification.** *Saccharomyces cerevisiae* Hsp82p, Aha1p, Aha1p$^{Δ11}$, Hch1p, Hch1p$^{Δ11}$, Cpr6p, and Sti1p were expressed in *Escherichia coli* strain BL21 (DE3) (New England Biolabs) from pET11d (Stratagene, La Jolla, CA, USA). Two versions of pET11d were used to express these proteins. Hsp82p, Aha1p, Aha1p$^{Δ11}$, Cpr6p, and Sti1p were expressed with N terminal 6xHis tags and Aha1p, Aha1p$^{Δ11}$, Aha1p$^N$, Aha1p$^{N-Δ11}$, Hch1p, and Hch1p$^{Δ11}$ were expressed with C terminal 6xHis tags. Cells were grown at 37 °C to an OD$_{600}$ of 0.8–1.0 and induced with 1 mM isopropyl-1-thio-D-galactopyranoside (IPTG). Cells expressing Hch1p, Hch1p$^{Δ11}$, Aha1p, Aha1p$^{Δ11}$, and Hsp82p were harvested after overnight growth at 30 °C. Cells expressing Cpr6p, and Sti1p were harvested after overnight growth at 37 °C. Cells were harvested by centrifugation and stored at −80 °C. Cells were resuspended in lysis buffer (25 mM NaH$_2$PO$_4$, pH 7.2, 500 mM NaCl, 1 mM MgCl$_2$, 20 mM Imidazole, 5 mM β-mercaptoethanol) and lysed using Avestin Emulsiflex C3 (Avestin, Ottawa, Ontario, Canada). Lysates were clarified by ultracentrifugation and His-tagged proteins were isolated on a HisTrap FF column using an AKTA Explorer FPLC (GE Healthcare). Isolated 6xHis-tagged proteins were then concentrated and further purified by size exclusion chromatography on a Superdex 200 (Hsp82p, Sti1p, Aha1p, Aha1p$^{Δ11}$, Hch1p, Hch1p$^{Δ11}$) or a Superdex 75 (Cpr6p) column (GE Healthcare)[7,30]. Purity of each protein preparation was > 95% as verified by coomassie-stained SDS–PAGE analysis.

**ThermoFluor thermal shift assay.** Thermal stability was used as an indicator of folding status of our C terminally 6xHis-tagged Aha1p, Aha1p$^{Δ11}$, Hch1p, and Hch1p$^{Δ11}$. We carried out a thermal shift assay with 5, 2, and 1 µM concentrations of each co-chaperone construct in triplicate using Sypro Orange (ThermoFisher Scientific)[42,43]. $T_m$ was calculated using area under the curve analysis in GraphPad Prism of plots of $ΔF/ΔT$ (change in fluorescence/change in temperature).

**In vitro ATPase assays.** ATPase assays were carried out using the enzyme coupled assay as previously described[7,17,30,40,41]. All reactions were carried out in triplicate, three times in 100 µL volumes using a 96-well plate. Absorbance at 340 nm was measured every minute for 90 min using a BioTek Synergy 4 and the path-length correction function. Average values of the experiments are shown with error expressed as standard error of the mean. The decrease in NADH absorbance at 340 nm was converted to micromoles of ATP using Beer's Law and then expressed as a function of time[40]. The final conditions of all the reactions are 25 mM Hepes (pH 7.2), 12.5 or 16 mM NaCl (in titration and cycling experiments, respectively), 5 mM MgCl$_2$, 1 mM DTT, 0.6 mM NADH, 2 mM ATP (co-chaperone titration and cycling experiments), 1 mM phosphoenol pyruvate (PEP), 2.5 µL of pyruvate kinase/lactate dehydrogenase (PK/LDH) (Sigma), and 5% DMSO. To correct for contaminating ATPase activity, identical reactions were quenched with 100 µM NVP-AUY922 and subtracted from unquenched reactions (DMSO control). In the titration experiments (Fig. 3), 1 µM of Hsp82p was added to reactions containing either 1, 2, 4, 8, 12, or 16 µM of Aha1p, Aha1p$^{Δ11}$, or 4 µM of Hsp82p was added to reactions containing either 4, 8, 16, or 32 µM Hch1p or Hch1p$^{Δ11}$. The ATPase assays were started by the addition of the regenerating system consisting of MgCl$_2$, DTT, NADH, ATP, PEP, PK/LDH. Fit lines were calculated according to the following equation $(Y = ((B_{max}{*}X)/(K_{app} + X)) + X_0)$[36]. In the ATP titration experiment (Fig. 7 and S1), 2 µM of Hsp82p was added to reactions containing 20 µM of either Aha1p or Aha1p$^{Δ11}$ or 4 µM of Hsp82p was added to reactions containing 40 µM Hch1p or Hch1p$^{Δ11}$. The regenerating system containing MgCl$_2$, DTT, NADH, PEP, and PK/LDH was added to the reactions and the reaction was started by the addition of either 12.5, 25, 50, 100, 200, 400, 800, or 1600 µM of

ATP. ATPase rates were analyzed with the Michaelis–Menten non-linear regression function in GraphPad Prism. In the co-chaperone switching ATPase experiments (Fig. 3e), 4 µM of indicated co-chaperones and buffers were added to the wells and allowed to mix for 10 min. 2 µM of Hsp82p was then added and allowed to mix for 10 min before starting the reaction through the addition of our regenerating system containing MgCl$_2$, DTT, NADH, ATP, PEP, PK/LDH. The ATPase rate shown in µM ATP hydrolyzed per minute per µM of Hsp82p (1/min).

**PET fluorescence experiments.** Double mutant S51C-A110W of yeast Hsp90 was synthesized using recombinant methods and modified with the fluorophore AttoOxa11 (AttoTec) as previously described[34]. Time-dependent fluorescence intensities were measured from Hsp90 samples in a quartz glass cuvette using a FP-6500 spectrofluorimeter (Jasco). Fluorescence was excited at 620 nm and emission intensities were recorded at a wavelength of 678 nm. Measurements were carried out at room temperature. Hsp90 samples were prepared in 50 mM phosphate, pH 7.5, with the ionic strength adjusted to 200 mM using potassium chloride, containing 10 mM MgCl$_2$ and 150 nM of AttoOxa11-labeled Hsp90 construct. 5 µM non-labeled wildtype Hsp90 protein was added and incubated at room temperature for 30 min to ensure that only one subunit in hetero-dimeric constructs carried the fluorophore. Aha-type co-chaperones were added at 20 µM concentration. Reactions were started by addition of 2 mM AMP-PNP. Fluorescence transients were analyzed by fitting exponential decay functions. Data of the Hsp90 sample without Aha-type co-chaperone required a bi-exponential function to fit them accurately, likely caused by molecular ground-state heterogeneity as discussed previously[34]. The reported time constant is the average time constant of two exponentials weighted by the respective, relative amplitudes. Data of Hsp90 samples pre-incubated with Aha-type co-chaperone fitted well to mono-exponential decay functions. Samples of Hsp90 in complex with Aha1p and Aha1p$^{Δ11}$ showed a burst phase kinetics after the addition of AMP-PNP, which was faster than the time resolution of the experimental setup. The burst phase was caused by strong acceleration of lid closure caused by Aha1p and Aha1p$^{Δ11}$. Values of τ are mean values of three measurements and errors are the standard deviation of these three measurements.

**Reporting Summary.** Further information on experimental design is available in the Nature Research Reporting Summary linked to this article.

**Data availability**
Data supporting the findings of this manuscript are available from the corresponding author upon reasonable request. A reporting summary for this Article is available as a Supplementary Information file. The source data underlying Figs. 3b–e, 4b, 4d, 5c, d, 6a, b, 7a, 9, and Supplementary Figs. 1 and 2 are provided as a Source Data file.

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

## Acknowledgements

We would like to thank Gary Eitzen (University of Alberta) for the anti-actin antibody. Work in the laboratory of P.L. was supported by funding from the Natural Sciences and Engineering Research Council of Canada (386803), the Canadian Institutes of Health Research (97870), and Alberta Innovates Health Solutions (200900500). R.M. is supported by fellowships from the Alberta Cancer Foundation and Alberta Innovates Health Solutions. Work in the laboratory of J.L.J. was supported by funding from the National Institutes of Health (P30 GM103324).

## Author contributions

R.M. designed and carried out experiments, and edited the manuscript. A.W. and J.S. carried out experiments. H.N. designed experiments and edited the manuscript. J.L.J. identified the S25P yeast strain and edited the manuscript. P.L. conceived the study, designed experiments, and wrote the manuscript.

## Additional information

**Competing interests:** The authors declare no competing interests.

