## [Peer Review File · Nature Communications]

Reviewers' Comments:

Reviewer #1:

Remarks to the Author:

The authors present a very nice investigation into the roles of a conserved motif within the Hsp90 cochaperones Hch1 and Aha1 thereby providing important new insight into how these cochaperones contribute the functional capacity of Hsp90. The authors provide clear and strong evidence that the conserved motif is necessary to modulate the ATPase activity of Hsp90.

However, limiting the work is the lack of evidence showing whether the motif is needed to have a functional effect on an Hsp90 client. Minimally, an examination of v-Src and Glucocorticoid Receptor activities should be assessed in the wild type and mutant backgrounds. As a minor point, a measure of the quality of the protein preparations used in the study should be provided (e.g., a Coomassie stained SDS-PAGE).

Reviewer #2:

Remarks to the Author:

Mercier et al investigate the role of a completely conserved N-terminal motif in the Hsp90 co-chaperone Aha1 and its relative Hch1. The authors generate two truncations mutants (Δ -11) of Aha1 and Hch1, removing the conserved motif, and compare the biochemical and cell biological activity of these mutants with wild type. Specifically, the relative stimulation of Hsp90's ATPase activity and the K_m (apparent affinity for the substrate ATP) are measured for the truncation and WT proteins under steady state conditions and by the application of Michaelis-Menten kinetics. The results show that the Δ -11 truncation in both Hch1 and Aha1 decreases both the acceleration of Hsp90's ATPase activity, and the apparent affinity for ATP. This is an interesting observation and is particularly enticing in light of the fact that there is no structural information for this conserved motif. Indeed the paper describing the crystal structure of the Hsp90-Aha1 complex investigated the deletion of this conserved motif (Meyer et al., 2004) but found its contribution to the ATPase activity to be less significant than determined in this work. It would be useful for the authors to discuss this further and suggest possible reasons for the discrepancy.

The authors also investigate the *in vivo* effects of the Δ -11 truncation in *S. cerevisiae*. Using ts mutations that are rescued by Hch1 and Aha1, and drug sensitivity, they find that the Δ -11 truncation mutants no longer rescue the phenotypes tested. These data show that the contribution of the conserved motif is therefore an essential component of Aha1 activity.

Several key elements of raw data have not been presented:

1. Are the truncated proteins folded? No evidence is presented for this basic control.
2. The raw data is not shown for the critical experiment in figure 6. This must be included - at least in the supplementary information
3. The reader cannot assess the fold changes in ATPase rate of K_m since the axes on the graphs are not sufficiently annotated and the values are not written anywhere.

I have two major concerns about the main conclusions presented in the Discussion:

4. The discussion states that Aha1 accelerates the ATPase activity therefore it is likely that it should slow the conformational changes that allow ADP release, but this is an incorrect assumption: the acceleration caused by Aha1, which is measured in this manuscript as an approximately 2.5-fold increase in V_{max} , includes both steps 1 and 3 in the scheme in Figure 7, and it does not necessarily follow that if V_{max} increases then step 3 decreases. In Michaelis-Menten kinetics V_{max} is related to K_m , which is composed of both the on and off rate of the substrate and k_{cat} , where k_{cat} includes both the rate of product formation and the rate of dissociation of the product from the enzyme.
5. K_m is dependent on k_{off} , k_{on} and k_{cat} The inverse of K_m is the sum of the inverse of k_2 (formation of the enzyme-product complex E.P) and k_3 (dissociation of P from E – in this case

ADP). As I understand the manuscript as presented, the authors main conclusion is that the N-terminal conserved motif in Aha1 and Hch1 control the rate of ADP release, i.e. essentially acting as a nucleotide exchange factor. However no conclusions about k_3 can be made by measuring K_m alone. Furthermore the inverse relationship between k_{cat} , k_2 and k_3 negates the possibility to relate the fold-change in V_{max} to k_3 because these aren't products but sums. These individual rate constants need to be measured, using e.g. stopped flow measurements, before the current conclusions are valid. For an example of the kind of detailed analysis required a recent example from the Hsp70 field has been published in NSMB by Rosam et al., 2018.

In summary the current conclusions, if correct, could indeed influence thinking in the field, with the suggestion that Aha1 is a nucleotide exchange factor, however these conclusions are not supported by the presented data.

Additional experiments that would bolster the significance and depth of the manuscript:

6. Since this conserved motif seems to contribute such an important activity to Aha1 & Hch1 activity in vivo, the authors should investigate this further. In particular how do these deletions affect client activation (e.g. vSrc) for Aha1 and Hch1?

Reviewer #3:

Remarks to the Author:

In their manuscript „The conserved NxNNWHW Motif in Aha1-type co-chaperones modulates the kinetics of Hsp90 ATPase stimulation and is essential for in vivo function“ the authors describe the characterization of an N-terminal conserved motif in Aha1-type cofactors. This motif is the strongest conserved sequence in these Hsp90 cofactors, but mostly uncharacterized to date. The authors find interesting results in respect to the ATPase regulation and in vivo function, but then do not support their hypotheses with further experimentation to get into the position to clearly state these points. While several findings in this study clearly are interesting and provide novel concepts, the study remains largely hypothetical due to missing approaches to support these concepts. This is summarized in the three main points to be discussed or supported.

1. The authors find biophysical evidence that the ADP release after hydrolysis could be rate-limiting under conditions of stimulated ATPase activity. This would be a new situation and assign an entirely new function to a cofactor of the Hsp90 system. Nevertheless the indications they present are not supported by an experiment that directly addresses the question of nucleotide-release and targets this point.

2. The authors present in vivo effects, which assign functions to Hch1 and Aha1 under in vivo conditions. The in vivo relevance of these cofactors has remained largely enigmatic so the presentation of in vivo conditions, under which a function for those two proteins becomes obvious, is a significant finding. The authors present these functions as growth inhibitory assays. It would be helpful to the reader, if at least in the discussion a section is included towards the molecular events, which could be causative for this slow growth or temperature dependent phenotype. As in two presented cases the phenotype is synthetic together with an Hsp82 or Hsc82 background mutation, the knowledge on the background mutation may present a chance to understand a physiological role for the involvement of Aha1 and Hch1 in cellular pathways.

3. Finally the authors present data that indicate that NVP-AUY922 binding to Hsp90 leads to hypersensitivity in cases, where Hch1 is functional, but not when the motif under investigation is deleted. This also is an interesting synthetic growth assay for Hch1, but here also the authors do not comment on their interpretation towards the cellular function of Hch1. Here the authors also mostly speculate on the combined action of this inhibitor and Hch1 on the ATPase function of Hsp90. Several of these aspects could be measured directly in the setups the authors already have in place, like ATPase assays with and without the inhibitor and Hch1.

In general the presented work has the potential to provide greatly wanted new insight into the function of Aha1 and Hch1 in yeast cells, but would require clarification or confirmation of one or

two of these points to get beyond the slightly far-fetching explanations towards an experimentally supported work with novel insights.

Reviewers' comments:

Reviewer #1 (Remarks to the Author):

The authors present a very nice investigation into the roles of a conserved motif within the Hsp90 cochaperones Hch1 and Aha1 thereby providing important new insight into how these cochaperones contribute the functional capacity of Hsp90. The authors provide clear and strong evidence that the conserved motif is necessary to modulate the ATPase activity of Hsp90. However, limiting the work is the lack of evidence showing whether the motif is needed to have a functional effect on an Hsp90 client. Minimally, an examination of v-Src and Glucocorticoid Receptor activities should be assessed in the wild type and mutant backgrounds. As a minor point, a measure of the quality of the protein preparations used in the study should be provided (e.g., a Coomassie stained SDS-PAGE).

Thank you for the kind comments and suggestions.

To address your first point, we have now carried out an analysis of v-src activation in the S25P mutant yeast background (revised Figure 5). Here we show that only wildtype Aha1p can promote v-src activity while the Aha1p^{Δ11} mutant cannot. This is consistent with the growth rescue experiments we included in the original manuscript.

To address your second comment, while we didn't mention it in the first draft of the manuscript, we do always confirm that all our protein preparations are >95% pure by coomassie staining. We don't usually include these gels as figures (but perhaps we should) but we have added this point to the methods section of the new manuscript. Importantly, our ATPase assays report only NVP-quenchable ATPase activity so even if there were ATPase contaminants in any of our proteins, that activity would not be reflected in our data. Additionally, we have included the results of direct measurements of the thermodynamic stability of our full length and Δ11 co-chaperone constructs in the text. We carried out ThermoFluor protein melting assays and found that there is no change in melting temperature for either Aha1p or Hch1p when the NxNNWHW motif is deleted - indicating that there is no disruption to folding of these co-chaperones when the NxNNWHW motif is deleted.

Reviewer #2 (Remarks to the Author):

Mercier et al investigate the role of a completely conserved N-terminal motif in the Hsp90 co-chaperone Aha1 and its relative Hch1. The authors generate two truncations mutants (delta-11) of Aha1 and Hch1, removing the conserved motif, and compare the biochemical and cell biological activity of these mutants with wild type. Specifically, the relative stimulation of Hsp90's ATPase activity and the Km (apparent affinity for the substrate ATP) are measured for the truncation and WT proteins under steady state conditions and by the application of Michaelis-Menten kinetics. The results show that the delta-11 truncation in both Hch1 and Aha1 decreases both the acceleration of Hsp90's ATPase activity, and the apparent affinity for ATP. This is an interesting observation and is particularly enticing in light of the fact that there is no structural information for this conserved motif. Indeed the paper describing the crystal structure of the Hsp90-Aha1

complex investigated the deletion of this conserved motif (Meyer et al., 2004) but found its contribution to the ATPase activity to be less significant than determined in this work. It would be useful for the authors to discuss this further and suggest possible reasons for the discrepancy.

We were struck by this as well - our initial experiments gave the same results as those you mentioned in the 2004 study. In the last two years we started to get a little uneasy with the location of the 6x His tag (the *N* terminus) when examining an *N* terminal truncation. To our knowledge, all previous work with Aha1p has used an *N* terminal His tag. We have now included a direct comparison of ATPase stimulation by *N* terminally and *C* terminally 6xHis tagged Aha1p (revised Figure 3B). These results show that the *N* terminal 6xHis tag impairs the catalytic/functional properties of full length Aha1p. This is reasonable because the *N* terminal 6xHis tag, which is of similar length as the motif, is directly linked to the conserved motif. We can thus explain the discrepancy of our results where we used the 6xHis tag at the non-perturbative, *C* terminal position with previous data from Meyer *et al.*, who used the tag at the *N* terminal position. From now on we will only be using *C* terminally 6xHis tagged co-chaperones and we hope this will guide future work in other groups as well.

The authors also investigate the *in vivo* effects of the delta-11 truncation in *S. cerevisiae*. Using ts mutations that are rescued by Hch1 and Aha1, and drug sensitivity, they find that the delta-11 truncation mutants no longer rescue the phenotypes tested. These data show that the contribution of the conserved motif is therefore an essential component of Aha1 activity.

Several key elements of raw data have not been presented:

1. Are the truncated proteins folded? No evidence is presented for this basic control.

We regret not addressing this directly in the original submission. We determined the melting temperature of each protein using a ThermoFluor protein melting assay with the full length and $\Delta 11$ co-chaperones. The T_m we observed ($n=9$) for both full length and truncated forms of Aha1p or Hch1p are identical. These values are noted in the text near Figure 3. We have also added the assay details to the materials and methods.

2. The raw data is not shown for the critical experiment in figure 6. This must be included - at least in the supplementary information

3. The reader cannot assess the fold changes in ATPase rate of K_M since the axes on the graphs are not sufficiently annotated and the values are not written anywhere.

We have prepared a table (Table 1) reporting the values for K_M . This appears near Figure 7 (which was Figure 6 in the original submission). We also added Supplemental Figure 1 showing the ATPase results used to determine K_M for ATP.

I have two major concerns about the main conclusions presented in the Discussion:

4. The discussion states that Aha1 accelerates the ATPase activity therefore it is likely that it should slow the conformational changes that allow ADP release, but this is an incorrect assumption: the acceleration caused by Aha1, which is measured in this manuscript as an approximately 2.5-fold increase in V_{max} , includes both steps 1 and 3 in the scheme in Figure 7,

and it does not necessarily follow that if V_{max} increases then step 3 decreases. In Michaelis-Menten kinetics V_{max} is related to K_M , which is composed of both the on and off rate of the substrate and k_{cat} , where k_{cat} includes both the rate of product formation and the rate of dissociation of the product from the enzyme.

5. K_M is dependent on k_{off} , k_{on} and k_{cat} The inverse of K_M is the sum of the inverse of k_2 (formation of the enzyme-product complex E.P) and k_3 (dissociation of P from E – in this case ADP). As I understand the manuscript as presented, the authors main conclusion is that the N-terminal conserved motif in Aha1 and Hch1 control the rate of ADP release, i.e. essentially acting as a nucleotide exchange factor. However no conclusions about k_3 can be made by measuring K_M alone. Furthermore the inverse relationship between k_{cat} , k_2 and k_3 negates the possibility to relate the fold-change in V_{max} to k_3 because these aren't products but sums. These individual rate constants need to be measured, using e.g. stopped flow measurements, before the current conclusions are valid. For an example of the kind of detailed analysis required a recent example from the Hsp70 field has been published in NSMB by Rosam et al., 2018.

We apologize for not precisely communicating our arguments in the discussion section. We did not mean to say that acceleration of ATPase activity of Hsp90 induced by Aha1 leads to slowing of conformational changes that allow ADP release. Our observation is that removal of the NxNNWHW motif in both Aha1p and Hch1p reduces co-chaperone-induced activation of ATPase, and at the same time significantly reduces the K_M , (i.e. the affinity for nucleotide is increased). This led us to conclude that the conserved N terminal motif has an important and possibly unusual role in activating the Hsp90 catalytic cycle. As pointed out by the reviewer, K_M depends on three rate constants which are k_{off} , k_{on} and k_{cat} . The latter is the ATP hydrolysis step in Hsp90 and the former ones are binding and dissociation of nucleotide. We agree with the reviewer that no safe conclusion can be made on the effect of the conserved N terminal motif on K_M without measuring the rate constants. In our revised manuscript, we now measured rate constants of conformational change associated with closure of the molecular clamp of Hsp90 induced by nucleotide binding (in collaboration with the laboratory of Hannes Neuweiler in Würzburg, Germany). Since it is known that in Hsp90 the rate constant of ATP hydrolysis (k_{cat}) is limited by the rate constant of conformational change (i.e. clamp closure; References 9, 34, 46, 53, and 54 in the revised manuscript), this can provide insight into the mechanistic role of the NxNNWHW motif. Interestingly, we found that presence or absence of the NxNNWHW motif both in Aha1p and Hch1p has no effect on the rate constant of conformational change associated with clamp closure and thus on the rate constant of ATP hydrolysis (k_{cat}). We concluded that the motif therefore has to affect nucleotide binding/dissociation (k_{on}/k_{off}). Since there is no reasonable explanation for how the NxNNWHW motif could possibly accelerate binding of ATP, which is diffusion-controlled, we argue that the motif alters k_{off} , (i.e. accelerates dissociation of ADP) thus affecting the speed of the entire catalytic cycle of Hsp90 at steady state. We agree with the reviewer that it is desirable to measure the rate constant of nucleotide dissociation directly. This however, is complicated by the fact that ADP binding does not trigger the conformational changes necessary for the acquisition of the catalytically active state. Such an experiment would (i) require a probe that directly signals the bound/unbound state of the nucleotide. And (ii), the probe must be able to measure the time course of ADP dissociation in the post-hydrolysis state of Hsp90. This is not possible in conventional bulk experiments because the chaperone in solution stochastically runs through the catalytic cycle. A single-molecule assay would need to be developed for this purpose, which is currently not at hand.

We revised the manuscript discussion section trying to clarify these arguments.

In summary the current conclusions, if correct, could indeed influence thinking in the field, with the suggestion that Aha1 is a nucleotide exchange factor, however these conclusions are not supported by the presented data.

Thank you for these encouraging comments. We have added additional data, in particular measurement of rate constants of Hsp90 conformational changes, which support our proposal that these co-chaperones do have exchange activity.

Additional experiments that would bolster the significance and depth of the manuscript:
6. Since this conserved motif seems to contribute such an important activity to Aha1 & Hch1 activity in vivo, the authors should investigate this further. In particular how do these deletions affect client activation (e.g. vSrc) for Aha1 and Hch1?

We have added a new panel to Figure 5 showing that Aha1p, but not Aha1p^{Δ11}, can promote v-src activation in the S25P yeast background. This shows that there is indeed a client activation defect linked to the loss of the NxNNWHW motif.

Reviewer #3 (Remarks to the Author):

In their manuscript „The conserved NxNNWHW Motif in Aha1-type co-chaperones modulates the kinetics of Hsp90 ATPase stimulation and is essential for in vivo function“ the authors describe the characterization of an N-terminal conserved motif in Aha1-type cofactors. This motif is the strongest conserved sequence in these Hsp90 cofactors, but mostly uncharacterized to date. The authors find interesting results in respect to the ATPase regulation and in vivo function, but then do not support their hypotheses with further experimentation to get into the position to clearly state these points. While several findings in this study clearly are interesting and provide novel concepts, the study remains largely hypothetical due to missing approaches to support these concepts. This is summarized in the three main points to be discussed or supported.

1. The authors find biophysical evidence that the ADP release after hydrolysis could be rate-limiting under conditions of stimulated ATPase activity. This would be a new situation and assign an entirely new function to a cofactor of the Hsp90 system. Nevertheless the indications they present are not supported by an experiment that directly addresses the question of nucleotide-release and targets this point.

We thank the reviewer for the encouraging comments. The criticism of insufficient experimental evidence for post-hydrolysis activity of Aha1 has also been made by Reviewer #2. We performed additional experiments, involving new collaborative work, and measured rate constants of conformational change of Hsp90 in complex with co-chaperones with and without the conserved N terminal motif. The results are detailed in our response to comments #4 and #5 of Reviewer #2 above. In essence, we found that the conserved, N terminal motif of Aha1 has no influence on the rate constant of closure of the Hsp90 molecular clamp. From this result we infer that the effect of the NxNNWHW motif lies in modulation of nucleotide binding.

2. The authors present *in vivo* effects, which assign functions to Hch1 and Aha1 under *in vivo* conditions. The *in vivo* relevance of these cofactors has remained largely enigmatic so the presentation of *in vivo* conditions, under which a function for those two proteins becomes obvious, is a significant finding. The authors present these functions as growth inhibitory assays. It would be helpful to the reader, if at least in the discussion a section is included towards the molecular events, which could be causative for this slow growth or temperature dependent phenotype. As in two presented cases the phenotype is synthetic together with an Hsp82 or Hsc82 background mutation, the knowledge on the background mutation may present a chance to understand a physiological role for the involvement of Aha1 and Hch1 in cellular pathways.

We have now carried out v-src activation assays in the S25P mutant yeast background with Aha1p and Aha1p^{Δ11} overexpression (Figure 5D). Here we show that Aha1p, but not Aha1p^{Δ11}, overexpression can promote v-src activity providing insight into the underlying biology of the growth phenotype we originally reported. We have also included an analysis of the ATPase activity of the S25P mutant of Hsp90 in Figure 6.

3. Finally the authors present data that indicate that NVP-AUY922 binding to Hsp90 leads to hypersensitivity in cases, where Hch1 is functional, but not when the motif under investigation is deleted. This also is an interesting synthetic growth assay for Hch1, but here also the authors do not comment on their interpretation towards the cellular function of Hch1. Here the authors also mostly speculate on the combined action of this inhibitor and Hch1 on the ATPase function of Hsp90. Several of these aspects could be measured directly in the setups the authors already have in place, like ATPase assays with and without the inhibitor and Hch1.

All of our ATPase assays report only Hsp90-inhibitor-quenchable ATPase activity. Each condition is carried out with and without NVP-AUY922. We have tried to measure differences in EC₅₀ for NVP-AUY922 in the presence of Hch1p but have not noted any difference. I think ATPase assays are an excellent way to measure apparent K_M for ATP but lack the sensitivity to accurately measure Hsp90 inhibitor binding. We show in Figure 7 that there is a large reduction in apparent affinity for nucleotide when Hch1p is present. We can only speculate if this means that access to the nucleotide binding site is altered *in vivo* (in the presence of client).

In general the presented work has the potential to provide greatly wanted new insight into the function of Aha1 and Hch1 in yeast cells, but would require clarification or confirmation of one or two of these points to get beyond the slightly far-fetching explanations towards an experimentally supported work with novel insights.

Reviewers' Comments:

Reviewer #2:

Remarks to the Author:

The revised manuscript has dealt well with the shortcomings identified in the original draft. The authors now present strong evidence for a novel role of Aha1-type chaperones as nucleotide exchange factors.

Reviewer #3:

Remarks to the Author:

The manuscript "The conserved NxNNWHW Motif in Aha1-type co-chaperones modulates the kinetics of Hsp90 ATPase stimulation and is essential for in vivo function" has been rewritten and substantially improved. The authors have responded to and satisfied all points raised by us and we now feel that this manuscript should be published as an important contribution to our understanding of the Hsp90 cofactors Hch1 and Aha1.